# Mind the Metrics: Patterns for Telemetry-Aware In-IDE AI Application Development using Model Context Protocol (MCP)

## Abstract

Modern AI-driven development environments are destined to evolve into observability-first platforms by integrating real-time telemetry and feedback loops directly into the developer workflow. This paper introduces telemetry-aware IDEs driven by Model Context Protocol (MCP), a new paradigm for building software. We articulate how an IDE (integrated development environment), enhanced with an MCP client/server, can unify prompt engineering with live metrics, traces, and evaluations to enable iterative optimization and robust monitoring. We present a progression of design patterns: from local large language model (LLM) coding with immediate metrics feedback, to continuous integration (CI) pipelines that automatically refine prompts, to autonomous agents that monitor and adapt prompts based on telemetry. Instead of focusing on any single optimizer, we emphasize a general architecture (exemplified by the Model Context Protocol and illustrated through a reference MCP server implementation) that consolidates prompt and agent telemetry for the future integration of various optimization techniques. We survey related work in prompt engineering, AI observability, and optimization (e.g., Prompts-as-Programs, DSPy's MIPRO, Microsoft's PromptWizard) to position this approach within the emerging AI developer experience. This theoretical systems perspective highlights new design affordances and workflows for AI-first software development, laying a foundation for future benchmarking and empirical studies on optimization in these environments.

## 1 Introduction

The rise of large language models and AI "copilot" systems has transformed software development, introducing AI agents into Integrated Development Environments (IDEs) to assist with coding, content generation, and decision-making (Hou et al., 2025). However, developing AI-driven applications poses new challenges: LLM-based components are non-deterministic, difficult to debug, and often behave in a non-transparent (black-box) manner. Traditional software engineering practices rely on observability (the ability to infer internal states of a system from its external outputs like logs, metrics, and traces) to understand and debug systems, and a specialized discipline, LLMOps (Large Language Model Operations), is emerging to address the unique lifecycle management challenges of LLMs (Solutions, 2023; Shi et al., 2024). We propose that AI-first IDEs should similarly be telemetry-aware, treating prompts and AI agent interactions with the same rigor as code, by integrating real-time telemetry (the collection and transmission of data such as evaluation metrics, trace logs, and performance signals from remote or internal system components for monitoring) into the development loop, a concept we term "Agent-Integrated Development Environment (AIDE)".

AI observability has recently emerged as a critical need for reliable deployment of LLM applications (Sergeyuk et al., 2024). Observability provides insight into an AI system's behavior by gathering runtime data (or telemetry) such as prompts, model responses, cost, and quality metrics. New platforms have been built to log and analyze LLMs, allowing teams to debug complex prompt chains, measure accuracy and hallucination rates, and assess how changes impact model outputs (Anonymous, 2025c). Yet, these capabilities are often external to the IDE and continuous integration process. We argue for observability-first design: making

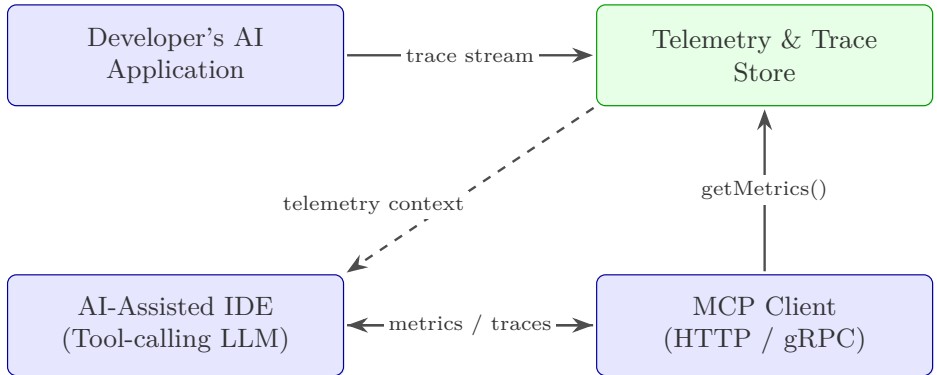

*Figure 1: Baseline telemetry workflow with the Model Context Protocol (MCP). The AI application streams traces directly into the telemetry store. An MCP client retrieves metrics/traces on demand. The AI-assisted IDE uses telemetry signals from the MCP client (via the diagonal dashed arrow), feeding its tool-calling LLM to refine prompts and code.*

telemetry and evaluations a first-class citizen within the IDE, so that developers (and the AI) can iterate on prompts and agent desgin with faster feedback, analogous to how they iteratively refine code using logs and tests.

This paper introduces telemetry awareness to an IDE to facilitate AI developement. This is inspired by the recently proposed Model Context Protocol (MCP), a standardized open-interface for connecting AI models with external tools and resources (Hou et al., 2025; Anthropic, 2024). In our vision, an MCP client/server acts as a unified broker of metrics (quantitative signals and traces), control (commands or adjustments to AI agents), and prompt data (prompt templates and versions). By interfacing the IDE with an MCP server, developers can seamlessly manage prompts, retrieve evaluation metrics, inspect traces of the LLM's reasoning, and could even send control instructions to running agents in a consistent and standardized manner. This transforms the IDE into an interactive AI observability dashboard as well as a coding environment.

To ground this paradigm, we describe a progression of design patterns that illustrate increasing integration of telemetry in the AI development workflow:

- Local Development with Metrics-in-the-Loop: Developers working in an AI-enhanced IDE receive real-time feedback from the LLM runs in the form of metrics and traces (including evaluations). The IDE can display evaluation results (e.g. error rates, token usage, latency, accuracy, hallucination) after each prompt or application execution and allow querying of trace logs. This immediate feedback loop helps identify flaws in prompts or agent logic early on.

- CI-Integrated Prompt Optimization: Telemetry-aware design extends to continuous integration. Prompt quality checks and automated optimizations run as part of CI pipelines, using stored traces and metrics to refine prompts or catch regressions. The MCP protocol provides a consistent API to fetch evaluation results and update prompts programmatically during testing and QA phases.

- Autonomous Monitoring Agents: In the most advanced pattern, autonomous monitoring agents leverage the telemetry stream to continuously watch an AI system in runtime and suggest improvements. These agents, which can be implemented as LLM-based evaluators or scripts, use the MCP interface to query interactions and metrics, then generate improved prompts or fine-tuning instructions without direct human intervention.

Crucially, we emphasize the architecture that makes these patterns possible. We detail how an open-source MCP client/server can unify prompt and agent telemetry across development and deployment environments, enabling future integration of optimizers. Rather than proposing a specific optimization algorithm, our focus is on the design affordances of this architecture, specifically how it creates a foundation upon which tools like DSPy's MIPRO or PromptWizard could plug in to automatically tune prompts using the rich data collected.

By decoupling the collection of metrics, traces and evaluations from the optimization logic, the MCP-based approach allows any number of optimization strategies (heuristic, learned, or hybrid) to be applied in a modular way.

This work is a theoretical and architectural insight paper. We do not present benchmark experiments or claim specific performance improvements. Instead, we identify emerging workflows and frameworks that point toward a new developer experience for AI applications. We cite related efforts in prompt engineering and LLM observability to show how the community is converging on this paradigm. Our hope is that this conceptual framework will inform and inspire more comprehensive evaluations in future work, where the impact of telemetry-aware development on model performance, developer productivity, and reliability can be systematically measured. However, we have made early progress on developing a prototype MCP client/server, applied to test-cases and have started on experimentation with IDE-assisted prompt optimization.

The rest of this paper is organized as follows. Section 2 reviews related work in AI-assisted IDEs, prompt optimization, and observability for LLM systems. Section 3 introduces the telemetry-aware design patterns in detail, illustrating each stage of the development lifecycle. Section 4 presents the MCP architecture and its implementation in an example system, describing how it unifies metrics, control signals, and prompts. Section 5 discusses the implications of this paradigm and outlines avenues for future research, including the integration of advanced optimizers. Finally, Section 6 concludes with a summary of key insights. An appendix provides supplementary materials and figures.

## 2 Background and Related Work

### 2.1 AI Integration in Development Environments

Integrated Development Environments (IDEs) have begun incorporating AI assistance to enhance programmer productivity and decision-making (Hou et al., 2025). For example, code completion and synthesis tools powered by LLMs (e.g., GitHub Copilot, Cursor, Zencoder's agentic solutions (Zencoder, 2024)) have become common. Recent studies survey the Human-AI experience in IDEs, highlighting the need to design appropriate user interfaces and interactions for these AI features (Wang et al., 2022). Key challenges include building trust in AI suggestions, ensuring readability of AI-generated code, and designing task-specific UI affordances for effective collaboration between the developer and the AI assistant (Anuyah et al., 2023; Wang et al., 2022). These findings underscore that integrating AI into development is not just a matter of model quality, but also of how information flows to the developer. Our work builds on this premise by channeling rich telemetry information into the IDE, thus giving developers deeper insight into the AI's behavior (addressing trust and transparency) and enabling more informed interventions.

Another trend is the emergence of frameworks where LLMs act as controllers or orchestrators in complex workflows. HuggingGPT is a prominent example, where an LLM (ChatGPT) manages calls to numerous task-specific models to solve multi-step AI tasks (Shen et al., 2023). In HuggingGPT, the LLM plans a sequence of subtasks, selects appropriate models from a model hub (HuggingFace), invokes them, and aggregates the results. This demonstrates an agentic use of LLMs that goes beyond single-turn prompting; the LLM essentially becomes a runtime decision-maker coordinating a pipeline. Such multi-component "AI programs" are powerful but notoriously difficult to debug or optimize. The work on AI Chains (Wu et al., 2022) highlighted these challenges, noting that in such chained systems, "Errors in one step can easily propagate to later steps, leading to unexpected final results. Without proper tools, it is hard to trace back to the root cause of errors" (Wu et al., 2022). Wu et al. specifically discuss the "brittleness" of these composite pipelines and the difficulty of "identifying the failing component" when, for example, a badly formatted prompt or an unexpected model output in an early step cascades into errors downstream (Wu et al., 2022). Indeed, errors can arise from prompt mis-specifications, model selection issues, or data passing between steps. This motivates first-class observability for LLM-driven workflows. To address this, the AI Chains system itself introduced an observability toolkit where its "interface displays the execution trace of an AI chain... For each step, the interface shows the input, output, and any intermediate results" (Wu et al., 2022). This fine-grained tracing is essential for making agentic or chain-based LLM systems debuggable and improvable. Participants in their study found this step-by-step trace "super helpful to see the input and output of each

step" making it easier to "find which step is wrong" and appreciated the ability to "replay the chain and see what happened at each step"(Wu et al., 2022). By instrumenting each step (e.g., logging model outputs, timestamps, intermediate prompts), developers can trace the chain of events and identify bottlenecks or failure points. Our proposed MCP-driven approach aligns with this need, as it provides a uniform way to capture and query traces from multi-agent or tool-augmented LLM systems. In fact, the value of capturing trace data for agent-based systems is already being recognized. For instance, modern LLM observability platforms can record full execution traces of agent frameworks like HuggingGPT or Google's Agent Development Kit, giving developers "deep visibility into agent behavior and orchestration" (Anonymous, 2025c).

## 2.2   Prompt Engineering and Optimization Methods

Prompt engineering has emerged as a critical discipline for aligning LLM behavior with user needs (Liu et al., 2023). Early prompt engineering often relied on manual craft and intuition (Liu et al., 2023; Perez et al., 2021; Reynolds & McDonell, 2021; Brown et al., 2020), but as applications scale, there is growing interest in automated prompt optimization (Zhou et al., 2023; Schnabel & Neville, 2024; Khattab et al., 2023). Two notable research directions are: (1) treating prompts as structured, modular programs, and (2) using AI feedback loops to refine prompts.

In the first category, prompts are viewed not as monolithic strings but as compositions of reusable components like instructions, examples, and context inserts. The "Prompts-as-Programs" concept formalizes this view (Schnabel & Neville, 2024), with Schnabel and Neville (2024) introducing Sammo, a framework for compile-time prompt optimization. In Sammo, a complex prompt (or "metaprompt") is represented as a call graph of components (e.g., functions to render different sections of the prompt). This structured representation allows systematic transformations akin to compiler optimizations, for example, pruning irrelevant instructions or tuning hyperparameters of prompt components (Schnabel & Neville, 2024). Sammo's approach was influenced by DSPy (Declarative Self-Improving Python), which also treats prompt engineering as programming. DSPy is an open-source framework that enables developers to define LLM application logic in a modular way and then optimize prompts through code-based experimentation. Rather than relying purely on trial-and-error, DSPy provides teleprompter algorithms (e.g., Cooperative, Bayesian optimizers) that iteratively adjust prompt instructions and few-shot examples to improve task performance (Khattab et al., 2023). The advantage of DSPy is that it integrates prompt tracking and tuning into a code workflow: one can programmatically log model outputs and evaluate them against assertions or examples, then let the optimizer propose better variants. This aligns closely with telemetry-aware development, where tracking essentially involves collecting metrics on prompt outcomes, and self-improvement uses these metrics to drive prompt updates.

In the second category, AI feedback loops, we see techniques like Microsoft's PromptWizard (Agarwal et al., 2024). PromptWizard is a fully automated framework where the LLM itself is employed to critique and refine prompts in an iterative loop. The process begins with an initial prompt (and possibly a few training examples) and proceeds as follows: the LLM generates multiple candidate prompt variants, tests them (by trying to solve the task or by using an evaluation function), and then analyzes its own outputs to suggest improvements. This "self-evolving" mechanism means the LLM alternates between being a solver and a critic, using each round's feedback to produce a better prompt in the next round. PromptWizard jointly optimizes the instruction part of the prompt and the choice of few-shot examples, even synthesizing new examples that expose weaknesses in the prompt. Notably, PromptWizard's design reflects a general template: provide a feedback signal, and let the model itself search for prompt improvements. The feedback can be as simple as task accuracy or a critique rubric, and the search is guided by the model's ability to introspect on errors. In essence, this approach also leverages telemetry (feedback metrics) to drive prompt updates, although the 'agent' acting on telemetry is the LLM itself.

The above approaches (Sammo, DSPy, PromptWizard) are complementary to our proposed paradigm. They supply algorithms and frameworks for optimizing prompts, while our MCP-based telemetry integration provides the necessary infrastructure to apply such optimizers continuously and contextually. In particular, an observability-first IDE could feed actual usage traces and evaluation metrics into a DSPy's MIPRO optimizer or PromptWizard loop. Prior work often used static datasets or fixed evaluation sets for optimization; by contrast, an MCP telemetry pipeline could enable optimizers to work with live data from real user interactions and test cases as they happen, closing the gap between prompt tuning and deployment behavior.

### 2.3 Telemetry and Observability for LLM Systems

Observability is a well-established concept in software reliability, referring to the ability to infer internal states of a system from its external outputs such as logs, metrics, and traces. For LLM-based applications, telemetry data includes prompt inputs, model outputs, intermediate reasoning steps (if available), timings, token usage, and user feedback, among other signals (Liang et al., 2023). Several recent works and tools underline the importance of LLM observability. On the conceptual side, Onose et al. (2024) define LLM observability as "the practice of gathering data (telemetry) while an LLM-powered system is running to analyze, assess, and enhance its performance" (Sergeyuk et al., 2024). This involves recording prompts and responses, tracing requests through any pipelines or chains, monitoring performance metrics (like latency and error rates), and evaluating output quality via automated or human feedback. The goal is to provide developers with visibility into the behavior of otherwise opaque AI components, thereby enabling debugging and iterative improvement.

Alongside the growth of LLM observability, the broader field of LLMOps (Large Language Model Operations) has rapidly evolved as an adaptation of MLOps principles tailored for the specific needs of LLMs (Solutions, 2023; Sinha et al., 2024). LLMOps encompasses the entire lifecycle of LLMs, from data preparation and model training through deployment, monitoring, and maintenance in production environments (Solutions, 2023; Pahune & Akhtar, 2025). This discipline addresses significant challenges inherent to large models, such as managing vast datasets, scaling computational resources (often requiring specialized hardware like GPUs/TPUs and parallelization techniques), and ensuring continuous model performance, which includes mitigating biases, detecting hallucinations, and countering model degradation over time (Solutions, 2023; Shi et al., 2024). Key best practices in LLMOps include robust data management, continuous model training and fine-tuning, designing scalable infrastructure, rigorous monitoring and observability, security and compliance, inference optimization, CI/CD for seamless updates, collaborative workflows, human-in-the-loop (HITL) mechanisms for quality control, and adherence to ethical considerations (Pahune & Akhtar, 2025). LLMOps emphasizes ensuring the stability, reliability, interpretability, and maintainability of models, often requiring real-time monitoring and proactive strategies to address issues promptly (Shi et al., 2024). Recent advancements further highlight the importance of HITL systems, adversarial testing for robustness, and mature CI/CD pipelines to enhance LLM accuracy and ensure scalable, reliable deployments (Pahune & Akhtar, 2025).

Multiple platforms have arisen to support LLM observability in practice, including various MCP implementations and other tools built on open standards such as OpenTelemetry (Smith & Doe, 2024; Blanco, 2023; Team, 2024; Schaffer, 2024). These platforms typically offer SDKs or APIs to instrument LLM calls in applications, sending trace data to a centralized dashboard. Common features are storing all prompt inputs and outputs, logging metadata (timestamps, model versions, prompt templates used), and attaching evaluations (e.g., automatically scoring model responses for correctness or toxicity via classifier models or LLM judges). For instance, one open-source LLM evaluation platform allows teams to "log, evaluate, and iterate" on LLM prompts and agent flows with a scalable backend (Anonymous, 2025c). A distinctive aspect of this platform is its focus on tracing and evaluation as core capabilities: it records the sequence of calls in an agent or chain, and it can run evaluation routines on each output (such as computing hallucination scores or comparing against reference answers). The platform must handle high-volume data from every LLM interaction and support analytical queries over this data. Technologies like OpenTelemetry, a standard for capturing and exporting trace data, have been extended to work with LLM-specific data (Smith & Doe, 2024; Blanco, 2023; Team, 2024; Schaffer, 2024; Jain, 2024; Ishan Jain, 2024; OpenTelemetry Community, 2024), allowing integration with existing APM (Application Performance Monitoring) tools and reflecting ongoing standardization efforts for AI/ML workloads.

The Model Context Protocol (MCP) has recently been proposed as a standardized way to integrate such observability and control across AI tools. Hou et al. (2025) introduce MCP as "a standardized interface designed to enable seamless interaction between AI models and external tools and resources, breaking down data silos and facilitating interoperability across diverse systems" (Hou et al., 2025). Anthropic also open-sourced their version of the Model Context Protocol in late 2024, aiming to create a universal standard for connecting AI systems with diverse data sources, thereby simplifying how AI systems access necessary data (Anthropic, 2024). Their work outlines the core components and workflow of MCP, including the concept

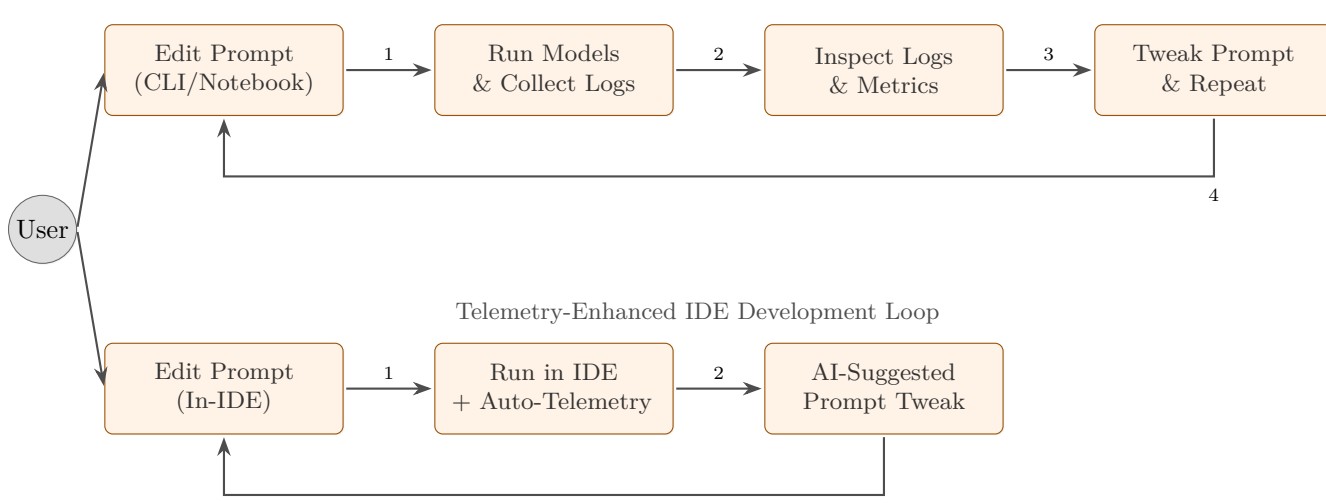

*Figure 2: Comparison between traditional prompt-engineering workflow (top) and MCP-enhanced IDE development loop (bottom). The traditional approach requires manual context switching between tools, while the telemetry-aware IDE integrates all steps within a unified environment, providing real-time feedback and AI-assisted prompt optimization.*

of MCP servers that manage the context for AI models (prompts, tools, state) through well-defined phases like creation, operation, and update. They also highlight industry adoption and use cases of MCP, indicating that companies are beginning to build infrastructure around this protocol for AI orchestration. Our use of the term MCP aligns with this vision, but we emphasize the "Metrics, Control, Prompt" interpretation as it relates to observability. In our telemetry-aware context, an MCP server is essentially an observability and control hub. It exposes APIs to log and query metrics/traces, to manage and version prompts, and to send control commands (for instance, instructing an agent to switch tools or trigger a reset) in a unified manner. The next section will illustrate how this MCP-based design underpins new development workflows.

# 3 Telemetry-Aware Design Patterns in AI Development

In this section, we outline three progressive design patterns for AI application development, each incorporating telemetry and feedback at increasing levels of automation. While various generative AI design patterns have been proposed (Koc, 2024), the feedback loop and direct interaction mechanisms for developers within many of these emerging LLM patterns remain an area requiring further clarification and development. These patterns are (1) Local metrics-in-the-loop coding, (2) CI-integrated prompt optimization, and (3) Autonomous monitoring agents. They correspond to stages in the development and deployment lifecycle, but the underlying principles are similar: use observability data to drive improvements in prompts or agent policies. We describe each pattern and provide examples of how an MCP-enabled IDE could facilitate the workflow.

## 3.1 1. Local Development with Metrics in the Loop

The first pattern involves the individual developer working within an IDE that provides real-time feedback about the AI's behavior. In a standard IDE, a developer writing code might rely on debug logs or a watch window to inspect program state after running the code. By analogy, in a telemetry-aware IDE, whenever the developer runs an AI-powered function (e.g., executes a prompt or tests an agent in a sandbox), the environment immediately surfaces telemetry insights: summary metrics and trace excerpts from that run.

Concretely, consider a developer iterating on a prompt for a question-answering agent. They write an initial prompt and execute it with some test query. A telemetry-aware IDE could automatically display information such as: the number of tokens in the prompt vs. completion, the model's response latency, and any evaluation results (e.g., whether the answer matches a ground truth if available, or a rating from a critique model). Moreover, the developer could inspect the trace of the agent's reasoning if the agent uses a chain-of-thought or tool use. For example, if the agent is supposed to call an API and then answer, the trace might show what API call was made and the result, enabling the developer to verify that the agent followed the correct steps.

The MCP integration allows the developer to ask the IDE (or rather, the LLM assistant in the IDE) questions about the telemetry: e.g. "From your trace logs, what are common sources of error?" or "Suggest prompt improvements based on the last 10 interactions.". The IDE forwards such queries to the MCP server, which has access to the stored traces and metrics of the project, and the LLM then composes an answer by analyzing that data. In the hypothetical case, the assistant identified that the conversation flow was slow (long response times) and users often provided certain inputs, then recommended adding explicit instructions in the prompt to structure those inputs. This kind of iterative loop – trial prompt → run → telemetry → revised prompt – can be done manually (with the developer reading metrics and adjusting) or in a human-in-the-loop manner (with the LLM suggesting adjustments based on telemetry and the developer accepting or tweaking them).

The benefit of local metrics-in-the-loop development is rapid feedback. Prompt engineering traditionally has a slow feedback cycle: one runs a prompt on a few examples and subjectively judges the outputs (Liu et al., 2023; Reynolds & McDonell, 2021). With integrated telemetry, the feedback becomes more objective and immediate, as the system can highlight quantitative issues (e.g., "The last 5 outputs had an average hallucination score of 0.7, which is high") and even point to specific trace examples that illustrate a problem. This accelerates the prompt refinement process. It also engenders a mindset shift – prompts and agent behaviors are not "magic" or static; they produce data that can be inspected and responded to. In essence, the developer is debugging prompts with logs, much as they would debug code with print statements.

Enabling this pattern requires the IDE to have access to a backend that accumulates and indexes telemetry for queries. The MCP client/server fulfills this role, acting as a repository of all prompts executed and their outcomes. A developer can retrieve, for instance, "the most recent trace with a high hallucination score" or "the output of the last conversation turn", via natural language queries that an MCP-enabled system interprets (Anonymous, 2025c;d). The IDE plugin translates these queries (possibly using a special syntax or an API call) to MCP requests. Under the hood, the server might store traces in a database and maintain metrics like token counts and evaluation scores per trace. When queried, it can filter and aggregate this information. The results are then fed back into the IDE's LLM assistant to generate a human-readable summary or recommendation. This seamless loop is what makes the observability-first IDE experience powerful: the developer is essentially conversing with both the LLM and the data about the LLM's past performance, all within one interface.

## 3.2  2. Continuous Integration with Telemetry-Guided Optimization

The second pattern extends telemetry-aware practices into the CI/CD (Continuous Integration/Continuous Deployment) pipeline. In modern software projects, CI is used to automatically run test suites, linters, and other analysis on every code change, catching regressions early. We propose that AI-centric projects incorporate prompt and agent evaluations in CI, enabled by the same telemetry infrastructure used in development.

A concrete example is integrating a suite of prompt test cases: for instance, a set of input queries with expected answer characteristics (not necessarily one correct answer, but perhaps certain constraints like "should cite a source" or "should not contain profanity"). Whenever a developer updates a prompt or changes the agent chain, the CI can run these test queries through the system (maybe using a fixed LLM checkpoint for consistency) and log the outputs to the MCP server. The MCP server, in turn, evaluates the outputs against the expectations using predefined evaluators or metrics (these could be simple regex checks, or more advanced LLM-based evaluators for coherence, etc.). The resulting metrics (like "pass/fail" counts for each test, or average scores) are then used to determine if the change introduced a regression. This is analogous to unit tests for code – here we have unit tests for prompts/agents. If a significant drop in performance is

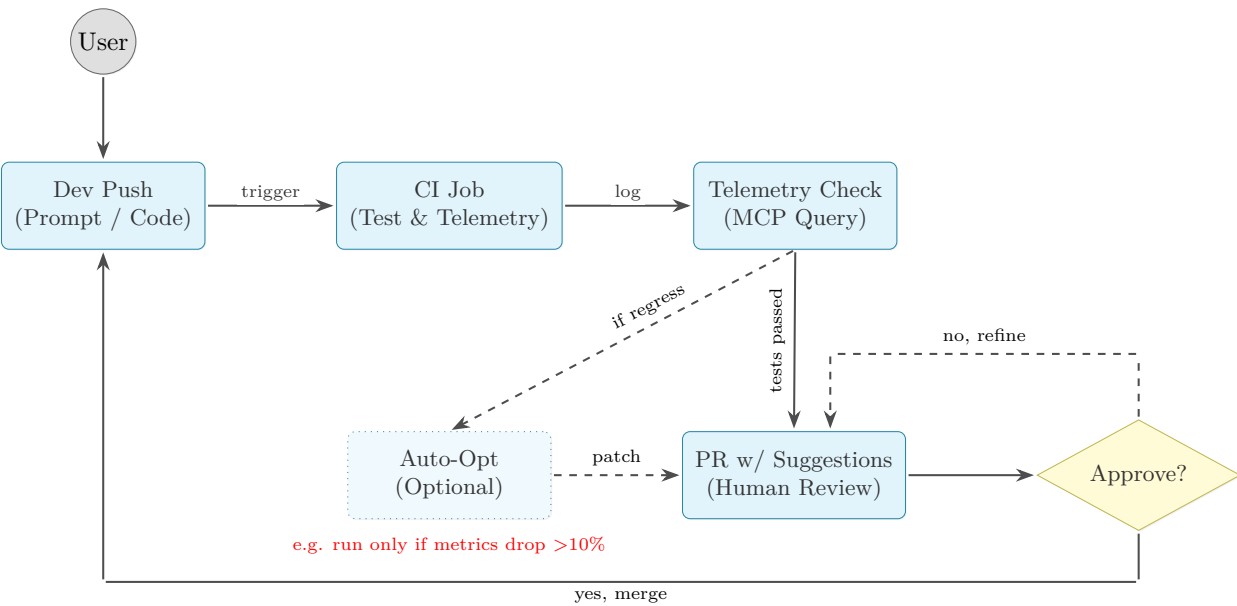

*Figure 3: CI/CD pipeline with MCP telemetry, user trigger, and clear approval flow. A user initiates the commit; telemetry checks drive optional optimization, and an approval decision routes to merge or further refinement.*

detected (say the average relevance score dropped by 10%), the CI can flag the build or even automatically revert the prompt change. The use of time-series data accessible through basic language queries can empower non-technical colleagues to define test cases and facilitate regression testing, for example, by monitoring for drift beyond a certain percentage point.

Telemetry plays a central role in this pattern by providing a historical baseline and context for each run. Because the server accumulates data over time, it can compare the current run's metrics to previous runs. For instance, it can answer: "Is the accuracy of the Q&A prompt in this commit within the expected range based on the last 10 commits?" If not, something might be wrong. Moreover, storing traces from CI runs means that when a failure occurs, developers can inspect the exact inputs and outputs that led to it, using the same IDE tooling described in Pattern 1. This tightens the integration between development and testing: rather than digging through log files on a CI server, a developer could simply query the MCP from their IDE (e.g., "Show me examples of where the new prompt failed the tests.") to retrieve the offending traces for review.

Another aspect is automated prompt optimization in CI. Since CI can execute code, it can also execute prompt optimization routines provided by frameworks like DSPy's MIPRO or PromptWizard (Khattab et al., 2023; Agarwal et al., 2024), as long as they expose an API or script. Imagine a scenario where a developer provides an initial version of a complex prompt, perhaps by roughly writing out sections and examples. Instead of expecting the developer to manually fine-tune it, the CI could trigger an optimization job that uses, say, DSPy's MIPRO algorithm or PromptWizard's self-refinement. This job would use the MCP data (or a static training set) as a basis to evaluate candidate prompts. The optimization might run for a few minutes and then output a refined prompt suggestion. The CI could then either automatically replace the prompt (which is risky and perhaps better for an offline process) or open a merge request for the developer to review the optimized prompt. Embedding prompt evaluation and optimization into CI/CD pipelines, as discussed, is increasingly recognized as a best practice within the broader systematization of LLMOps (Pahune & Akhtar, 2025). Essentially, the continuous part of CI would include continuously searching for better prompts whenever changes occur.

There are early signs of this approach. For example, DSPy's philosophy is to integrate prompt tuning into the development pipeline programmatically. One could incorporate a DSPy "program" in the repository that defines the prompt and an optimization objective; the CI then runs that program to update the prompt.

Microsoft's PromptWizard, while presented as an interactive research tool, could be adapted to run headless in CI to improve a prompt overnight. The key enabler is having a reliable evaluation metric or suite, which the telemetry system provides. If the project has accumulated real usage traces and labeled outcomes, those can serve as a testbed for optimization in CI (which is superior to generic benchmarks because it reflects the actual use case).

In summary, telemetry-aware CI means treating prompt/agent quality as a continuously tested property of the system. The MCP protocol provides the glue: it ensures that both the IDE and CI jobs speak the same language to log and retrieve prompt executions and their evaluations. A prompt tested in CI with certain metrics will have those metrics stored via MCP, and the developer can later query or visualize them in the IDE. Conversely, issues discovered during local development (and their fixes) can be codified as CI tests to prevent regressions. This synergy leads to a more robust development lifecycle, where improvements and bug fixes in prompts are tracked and validated just like code changes.

### 3.3  3. Autonomous Monitoring and Self-Improvement Agents

The third pattern is forward-looking: it envisions deployed AI systems that include autonomous agents monitoring other agents. Once an application is running in production (e.g., a deployed chatbot or an agent-based workflow serving users), telemetry can be used not only by human developers but also by automated "watchers" that ensure the system behaves optimally. We refer to these as autonomous monitor agents.

Consider a complex multi-agent system, for instance, a customer support bot that has multiple sub-agents for different tasks. This system produces a continuous stream of interactions and outcomes. An autonomous monitor agent could be implemented (possibly as an LLM itself or a traditional program) which subscribes to this stream via the MCP interface. This agent might periodically query the MCP server for recent traces with certain characteristics, for example: "find conversations in the last hour where the user was unhappy (negative feedback) or the agent's answer had a low confidence score." Given those problematic traces, the monitor agent could attempt to diagnose the issue. If it's implemented as an LLM prompt, it might take those traces and analyze them: "What went wrong in these interactions? Is there a pattern?" Suppose it finds that many failures involve a particular tool the agent is using incorrectly. The monitor could then formulate a suggested fix, perhaps "When using the database lookup tool, ensure to check for null results," which could translate to a prompt adjustment or a minor code modification in the agent's logic.

A more concrete example involves OpenAI's function-calling API, which allows tools to be invoked by the model (OpenAI, 2023). Imagine an agent that uses function calls to retrieve data. A monitor agent could watch function call traces via MCP and notice if a certain function call often returns errors. It could then autonomously create a pull request or a patch to the agent's prompt, adding an instruction like "If the database query fails, try an alternative query or apologize to the user." This patch would go through the normal CI process which, thanks to telemetry integration, can evaluate if the patch helps. In a sense, the monitor agent is acting as a specialized developer that continuously fine-tunes the system using evidence from telemetry. This also means additional logging and testing may not be required within a given AI application, as existing LLM-specific telemetry solutions can be leveraged.

While fully autonomous self-improving systems are still experimental, components of this vision are taking shape. Microsoft's PromptWizard already demonstrates an LLM refining prompts based on feedback in a loop, which can be seen as a micro-scale "self-healing" mechanism for a prompt. There are also research efforts on meta-prompts where one LLM monitors another's outputs for correctness or safety, often called an "evaluator" or "judge" model. Our paradigm can incorporate such evaluator models as first-class agents connected via MCP. For example, a hallucination watchdog LLM could be subscribed to each new answer produced by the main model. If it detects a likely hallucination (perhaps by cross-checking facts), it could trigger an intervention, such as logging an alert via MCP or even instructing the main model to provide sources.

The architectural advantage of MCP here is unification and access control. A monitor agent needs a lot of data to be effective – potentially a feed of all interactions. MCP can provide a filtered stream (via an API or event subscription) of telemetry data to authorized monitor agents. Because MCP manages prompts and

context, these monitor agents could also use it to update prompts or settings. This is the Control aspect of MCP: beyond passive observation, the protocol could allow certain clients to make changes (e.g., deploy a new prompt version or adjust a parameter) in a controlled manner. In practice, such control actions would be gated by policies or human review for safety. However, even without full autonomy, a monitor agent could at least propose changes. It might open a ticket or notify a developer with a suggested improvement and relevant evidence from logs.

Autonomous monitoring agents lead the ultimate goal of telemetry-aware design: a system that not only observes itself but improves itself over time (in an evolutionary manner). Achieving this in production reliably will require advances in ensuring the quality of the agent's suggestions (so that it doesn't introduce new problems) and in governance (to avoid a scenario where an autonomous agent makes inappropriate changes). That said, having the MCP infrastructure in place greatly simplifies prototyping such agents, because they can be developed and tested in the same way any client uses the MCP API. One could develop a monitor agent offline, feed it past telemetry data from MCP (e.g., logs from last week) to see if it correctly identifies issues, and then gradually deploy it to live monitoring. This incremental path is similar to how one might deploy a new microservice – starting in shadow mode, then giving it more responsibilities. The key difference is that the monitor agent's domain is the AI's own behavior.

## 4    Architecture: The MCP Server for Unified Telemetry and Control

Central to the above design patterns is the MCP server – the component that enables IDEs, CI pipelines, and agents to all share a common view of prompts, metrics, and traces. We now describe the architecture and capabilities of the MCP server in more detail, using a reference implementation (Anonymous, 2025a;b) as an illustrative example. The goal is to show how the MCP server serves as the unifying backbone that links development (IDE), testing (CI), and runtime monitoring in an AI-first application.

Unified Interface and Transport: The MCP server exposes a consistent API for clients to perform operations such as logging a trace, querying stored traces, saving or retrieving prompt templates, listing projects or experiments, and fetching aggregate metrics through cookbook-style docs and examples that the IDE can use as examples to enable logging, metrics, and telemetry. Importantly, it is designed to be accessible from different environments. For instance, the reference MCP server supports multiple transport mechanisms so it can integrate with various IDEs (Cursor, VS Code plugins, etc.) and also with non-UI clients (Anonymous, 2025c;d). In an IDE like Cursor, a developer can add the MCP server as a connection (specifying an API key and endpoint). Thereafter, the IDE's AI assistant can use this channel to communicate with the server for any telemetry-related query. The choice of standard protocols means the integration is relatively lightweight and doesn't require deep changes in the IDE, a key engineering decision to allow quick adoption.

Prompt Management: One core function of MCP is managing prompt versions and libraries. The server can store prompts (e.g., system prompts or few-shot example sets) keyed by names or IDs. Through the MCP API, a user or agent can list available prompts, fetch the latest version of a prompt, or save a new version (Anonymous, 2025d). This is extremely useful for keeping track of prompt iterations. Hypothetical as an example, when a developer accepts an optimized prompt recommendation, the IDE could save that new prompt to the MCP server with a version tag or comment. Now the CI pipeline and other team members have access to it. Conversely, if the CI's automated optimizer finds a better prompt, it could update the repository and also call the MCP API to record this prompt. Storing prompts centrally also enables prompt reuse and templating across projects – teams can build a library of tested prompts for common tasks (summarization, Q&A, etc.), queryable via MCP. This concept, enabling the creation of libraries of tested and reusable prompts, is also related to ideas like "Prompt Baking" (Bhargava et al., 2024).

Trace Logging and Analysis: The MCP server logs traces for each interaction or sequence of interactions as directed by the application. A trace typically includes the prompt used (or chain of prompts in an agent), the model's output, any tool calls and their results (for agent scenarios), and metadata like timestamps, model identity, and evaluation scores. The implementation provides methods to log various types of traces such as basic LLM calls, multi-step agent traces, conversation logs, and even distributed traces that tie together multi-component processes (Anonymous, 2025c). This flexibility suggests that MCP is meant to capture not only single-turn prompt-response pairs but also complex graphs of events. For analysis, the MCP server

can be queried with filters (e.g., by time range, by specific prompt or scenario, or by an evaluation metric threshold). As described earlier, the server can answer questions like "How many traces have been logged to the 'demo' project?" or "What is the latest trace's output?" directly (Anonymous, 2025d). These queries abstract away the underlying database queries. The user (or LLM agent) can simply ask in natural language via the IDE, and the MCP handles retrieving the info to feed into the LLM's answer. Additionally, the server can perform simple analyses like counting occurrences of events or computing average token usage, which can then be presented as part of telemetry summaries.

It's worth noting that because MCP can store evaluation results alongside traces, the queries can combine them. For example, "search for the most recent traces with high hallucination scores in project X" (Anonymous, 2025d) is a query that relies on an evaluation metric (hallucination score) being logged for each trace. The MCP server thus functions as a search engine over both raw interaction data and derived metrics. This capability exemplifies why unifying these concerns (Metrics, Prompt, and Control) is powerful: one can correlate prompt versions with metrics easily, or find problematic outputs and directly retrieve the prompts that led to them.

Metrics and Project Dashboard: In addition to individual traces, the MCP server maintains aggregate project metrics. A project in this context might correspond to an application or an experiment (e.g., "Banking Customer Support Chatbot v1"). Metrics could include things like average response time, success rate of queries, and token consumption over time. These can be fetched via the API for building dashboards or triggering alerts. The MCP server documentation mentions "fetching project metrics" as part of the interface (Anonymous, 2025d). One could imagine a CI pipeline querying these metrics after a load test to decide if the latest deployment meets performance targets. Likewise, a monitor agent might ask for a trend (e.g., "has the satisfaction score improved this week?"), which the MCP might answer by providing stats if such a metric is tracked. By centralizing metrics, MCP avoids the duplication of instrumentation code; every client, be it the IDE, CI, or an external monitoring service, taps into the same store.

Control and Tooling Integration: The 'Control' aspect of the MCP telemetry pattern is perhaps the most forward-looking. In the current implementation, control is implicit in the sense that one can save prompts or initiate evaluations via the MCP API, actions that cause changes. However, one can envision more explicit control messages. For instance, a command could be sent to switch the active prompt for an agent to a different version for A/B testing prompts. Or a control signal could pause an agent if a certain metric threshold is exceeded, like halting an agent that has gone into an infinite loop, based on trace analysis. While these are not detailed in the documentation we have, the architecture allows insertion of such commands since the MCP server sits between the IDE (or other clients) and the model execution environment.

A practical example of control could be the Agent Development Kit (ADK) integration mentioned earlier. Through MCP integrating with a given agentic framework, developers are allowed to start/stop agents and get full trace visibility from the IDE (Anonymous, 2025c;d). This implies a control channel where the IDE can send an instruction like "launch agent with config X and feed outputs to server." Similarly, LLMs with function calling, such as OpenAI's, can be monitored and perhaps controlled (e.g., disabling a function/tool if it leads to errors) via such a protocol.

From a systems perspective, the MCP server can be seen as middleware in the LLM AI toolchain. It doesn't replace the LLM or the application's code; instead, it instruments and mediates. By doing so, it decouples optimization and analysis from the core application logic. This decoupling is essential for plugging in external optimizers or validators. For example, one could run a separate process that periodically queries MCP for recent performance metrics and, if certain conditions are met, calls an optimizer library (like a DSPy MIPRO optimizer) to generate a better prompt, which it then submits back to MCP and possibly to a repository. The application using the prompt might then be notified or can pull the updated prompt at its next startup. Because MCP provides a version-controlled prompt store, rolling back is also straightforward if an optimization misfires, as one can revert to an earlier prompt version.

Security and permissions are also an architectural consideration (though beyond our scope to detail): since MCP centralizes access to potentially sensitive data (model inputs/outputs, user queries, etc.), it needs access control. The MCP design discussed by Hou et al. (2025) suggests phases like creation and update include considerations of privacy and security. In practice, API keys, role-based permissions (who can query vs. who

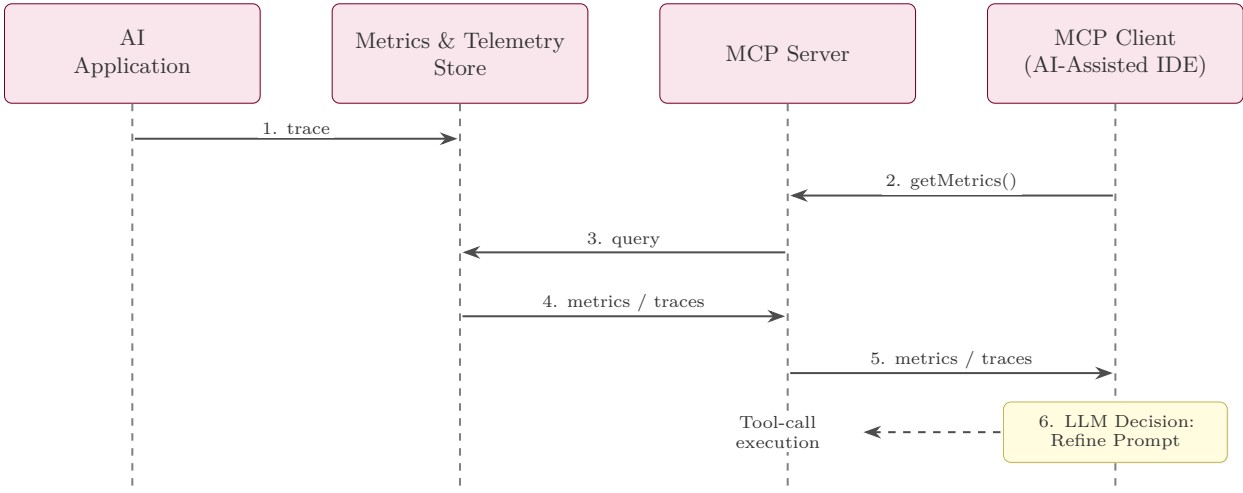

*Figure 4: Telemetry round-trip message sequence. (1) Application streams a trace to Metrics & Telemetry Store (Opik). (2) The IDE/LLM requests metrics via the MCP client, which queries Opik Store (3). (4)-(5) Metrics/traces flow back to the IDE/LLM, where (6) a local LLM tool-call can refine prompts based on the received telemetry.*

can modify prompts) would be part of a robust MCP server deployment. For an open-source MCP server implementation, developers can self-host it and thus control their data, or use a cloud service with guarantees.

In summary, the MCP server architecture provides:

- A single source of truth for prompts and telemetry data.

- Standardized queries for analysis and debugging.

- Interoperability between different stages of development (IDE, CI, production monitoring).

- Extensibility for future integration of optimizers and automated agents.

By emphasizing architecture over any single algorithm, we allow the ecosystem to evolve. Today's state-of-the-art optimizers (e.g., MIPROv2 in DSPy (Khattab et al., 2023) or PromptWizard (Agarwal et al., 2024)) can be tomorrow's legacy, as new ones will emerge. But with an MCP-based design, the AI development workflow does not need to be reinvented for each new optimizer. The optimizers can simply hook into the same telemetry streams and update the same prompt repositories. This unification is analogous to how the rise of logging frameworks and A/B testing platforms in web development allowed many different analytics or optimization modules to plug in without modifying the core app. We foresee the MCP paradigm playing a similar enabling role for AI-first software.

## 5 Discussion and Future Directions

The telemetry-aware IDE paradigm outlined here is in an early stage of adoption, but it signals a broader shift in how we conceive of AI development workflows. In this section, we discuss some implications of this paradigm, potential challenges, and future research directions that emerge from our proposal.

Changing Developer Roles and Skills: If IDEs become observability-rich AI development platforms, the skill set required for prompt/agent engineering may evolve. Developers will not only write prompts or glue code but also define metrics, interpret telemetry dashboards, and perhaps write meta-prompts to query their own systems. The workflow begins to resemble a dialogue with the AI system: you build the system, then ask the system about itself to improve it. This could lower the barrier for debugging AI behavior, since you can ask in natural language for summaries of failures, but it also means developers must think in terms

of experiments and data analysis. Future studies could examine how developers adapt to these tools. For example, does seeing quantitative metrics in the IDE lead to more objective prompt tuning decisions? Does having a conversational interface to logs make debugging more accessible to non-programmers? These human factors questions will be important for tool designers.

Benchmarking the Paradigm: We intentionally avoided any performance claims in this paper, focusing on conceptual and architectural benefits. However, a logical next step is to empirically evaluate telemetry-aware workflows. One could set up user studies where some developers use a telemetry-integrated IDE and others use a standard IDE to accomplish the same AI task, measuring differences in success rates or time. On the system side, one could benchmark how quickly an automated optimizer converges when it has access to live telemetry versus when it's run offline on a static dataset. The expectation is that tighter feedback loops yield faster iteration and more robust solutions, but this needs validation. Particularly interesting would be measuring the impact of autonomous monitoring agents – can they actually catch and fix issues faster than human operators? To enable such research, common evaluation tasks and datasets for 'AI self-debugging' could be developed (e.g., a suite of known prompt pitfalls that an autonomous agent should detect and correct).

Integration with LLM Evaluation Research: Our paradigm intertwines with the complex field of LLM evaluation – using LLMs as judges or designing new metrics for quality. The telemetry platform provides the data for evaluation, but what metrics to compute and how to interpret them is a whole challenge on its own. An open question is: Which telemetry signals are most indicative of real performance issues? It could be the frequency of user corrections, or latency spikes, or certain embeddings drifting – the MCP can log it all, but someone has to decide what triggers an optimization. Research into metric design (especially composite metrics for things like "helpfulness" or "harmlessness" of an agent) will directly feed into how effective telemetry-aware development can be. If the metrics are poor, developers could be misled by the telemetry (optimizing for the wrong thing). This calls for careful design of evaluation criteria and possibly interactive tuning of those criteria.

Orchestration and Interoperability: As multiple tools (IDE, CI, monitoring agents, optimizers) interconnect via MCP, orchestration becomes important. There may be scenarios of conflicting suggestions, for example, an automated optimizer proposes a prompt change that the monitoring agent disagrees with based on recent user feedback. How do we reconcile different "advisors" to the development process? One approach could be a central policy or meta-agent that takes input from various sources (human developers, CI tests, monitor agents) and makes decisions. This starts to look like an AI development orchestration layer, which could be another LLM or a rule-based system. We are essentially building an ecosystem where parts of the software development lifecycle are AI-driven (e.g., tests, analysis, optimization), and these need coordination. Establishing standards, possibly extensions of MCP, for how these components communicate their suggestions and outcomes will be useful. The current MCP focuses on context and telemetry; future iterations might include a schema for "proposed changes" or "hypotheses" that agents can submit.

Limitations and Risks: A discussion of this new paradigm must acknowledge potential pitfalls. One risk is over-reliance on automated feedback, as developers might trust the AI's self-evaluation too much. If an LLM says, "I have improved the prompt and errors are down by 50%", a developer might be tempted to accept that at face value. But perhaps the evaluation metric was incomplete. Ensuring a human-in-the-loop for validation, especially for changes that go to production, is advisable at the current state of technology. Another risk is data privacy and security: telemetry data can include sensitive user inputs or model outputs. Centralizing it in an MCP server means one must secure that server and comply with data handling norms, such as anonymization or encryption. The more we integrate these systems from IDE to production, the higher the stakes if there's a leak or misuse. Techniques like redaction of sensitive info in traces or limiting access scope will be important to implement alongside the functional features. Furthermore, prompt optimization algorithms can be time-consuming, and the associated latency might detract from the developer experience at present.

Future Optimizer Integration: One of our core motivations was to pave the way for integrating any advanced optimizer into the workflow without friction. We touched on how DSPy (Khattab et al., 2023) or PromptWizard (Agarwal et al., 2024) could plug in. Looking ahead, optimizers might evolve to use reinforcement learning

or advanced search that runs continuously. For a more detailed example of such a background service, see Appendix A.

Community and Standards: Finally, to truly realize telemetry-aware AI development at large, community standards and best practices should form. The Model Context Protocol itself is a candidate for standardization (Hou et al., 2025; Anthropic, 2024). If multiple vendors and open-source projects adopt a common protocol, IDEs and tools can interoperate. For instance, one could use VS Code with an MCP plugin that works with any compliant telemetry and metrics server. This would echo how LSP (Language Server Protocol) standardized language tooling integration in IDEs. We might see an LLM Telemetry Protocol as an extension of MCP focusing on observability data interchange. Additionally, sharing of telemetry datasets (anonymized) could spur innovation: imagine a public repository of agent trace logs and outcomes that researchers use to test new evaluation metrics or optimization methods. We encourage the community to explore such collaborative efforts, as the challenges of aligning AI behavior are too vast for siloed solutions.

## 6 Conclusion

We have presented a vision for observability-first AI IDEs centered around a unified Metrics, Control, Prompt (MCP) pattern, facilitated by the Model Context Protocol. By weaving real-time telemetry and feedback loops into the fabric of development tools, this paradigm enables a new level of insight and iterative improvement in AI application development. We described design patterns ranging from immediate in-IDE metrics feedback to fully autonomous self-optimization agents, all supported by an architectural backbone that treats prompts and their telemetry as first-class artifacts.

This approach reframes prompt engineering and agent design from a static art into a dynamic, data-driven process. It draws on ideas from prompt optimization research (such as treating prompts as programs and using AI critics) and from traditional software observability (like instrumentation and monitoring), unifying them in a practical development workflow. The MCP server exemplifies how such unification can be achieved: by providing a common interface for logging, querying, and updating AI context, it bridges the gap between development, testing, and production monitoring.

We emphasize that this is a theoretical and conceptual contribution. The potential benefits, such as faster debugging of AI issues, continuous improvement of model behavior, and safer deployments through constant evaluation, are compelling, but realizing them will require further experimentation and user studies. As a next step, the insights from this paper can inform the implementation of telemetry-aware features in popular IDEs and AI development frameworks. We envision that in the near future, asking your IDE "how is my AI model performing?" will be as natural as checking your code for compile errors. When that happens, the journey from debug logs to dynamic prompts will have come full circle, fulfilling the promise of truly intelligent development environments.

Ultimately, telemetry-aware AI-first IDEs represent a convergence of software engineering and AI engineering practices. Embracing this paradigm could lead to more robust, transparent, and adaptable AI systems – systems that not only learn from data but also learn from their own operation, in partnership with human developers. We hope this work provides a foundation for that evolution and inspires further innovation in building the next generation of AI development tools.

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

## A  Elaboration on Future Optimizer Integration Concepts

An interesting idea is a background optimizer service that constantly runs experiments in parallel (using spare compute, perhaps) to try and improve prompts or model parameters. MCP could feed it a stream of random real queries (privacy permitting) and the optimizer could test slight prompt variations to see if it yields better answers, all logged under a separate project or sandbox. If it discovers something better, it could alert developers. This resembles AutoML (automated model tuning) but applied to prompts/policies. With the telemetry infra, such experimental branches can be safely conducted without affecting users until confirmed. We anticipate research on online prompt optimization algorithms that use live traffic in a safe way (multi-armed bandit approaches, for example, to slowly route a small percentage of requests to a candidate prompt to gather metrics). The MCP's unified logging would capture both control and experimental groups for analysis.

