# OpenReview forum: "Mind the Metrics: Patterns for Telemetry-Aware In-IDE AI Application Development using Model Context Protocol (MCP)"
_TMLR — Rejected by TMLR_

### Review · Reviewer_MDSP · 2025-06-30

**Summary Of Contributions:**

This work introduces ​​telemetry-aware integrated development environments (IDEs)​​ as a conceptually novel paradigm for AI application development, leveraging the ​​Model Context Protocol (MCP)​​ to unify real-time observability metrics, prompt management, and control signals within developer workflows. The core vision transforms IDEs into observability-first platforms where developers iteratively refine prompts and AI agent logic using immediate feedback from traces, performance metrics, and evaluations, which are enabled by an MCP server acting as a centralized telemetry broker. This work proposes three progressive design patterns: (1) ​​local metrics-in-the-loop coding​​ for real-time debugging; (2) ​​CI-integrated prompt optimization​​ to automate quality control; and (3) ​​autonomous monitoring agents​​ that self-adapt using runtime telemetry. By architecturally decoupling telemetry collection (via MCP) from optimization algorithms, this approach establishes a foundation for future AI engineering tools to iteratively enhance system reliability and performance while bridging gaps between development, testing, and production monitoring. This work remains largely theoretical, offers insights, and lacks empirical validation.

**Audience:**

No

**Broader Impact Concerns:**

Since this work is closely related to AI application development, there are clear concerns regarding privacy and security. Although the authors have addressed some of these issues within the paper, it is advisable to present a dedicated section, e.g., a Broader Impact Statement, to discuss these matters more thoroughly.

**Claims And Evidence:**

No

**Requested Changes:**

Could the author provide a demo case that follows the workflow they have proposed to demonstrate the superiority?

**Strengths And Weaknesses:**

From my own perspective, this paper is interesting and of importance to AI application development in practice.

Strengths:
- offer a novel paradigm for AI application development;
- propose a workflow that demonstrates a coherent evolution of telemetry integration, with relevance to real-world AI development cycles;
- combine ideas from prompt optimization, observability, and CI/CD into a single, actionable workflow.

Weaknesses:
- Actually, I am not sure whether this paper (which provides insights but no experimental results at all) meets the scope of TMLR, which is the most obvious weakness.

---

### Review · Reviewer_LDUu · 2025-07-04

**Summary Of Contributions:**

This paper proposes a paradigm for "telemetry-aware IDEs" that integrate real-time metrics, traces, and evaluations into AI application development workflows using the Model Context Protocol (MCP). The authors present three progressive design patterns: local development with metrics-in-the-loop, CI-integrated prompt optimization, and autonomous monitoring agents.

**Audience:**

No

**Claims And Evidence:**

No

**Requested Changes:**

I asked some questions and changes in the strengths and weaknesses section.

**Strengths And Weaknesses:**

## Strengths

- The paper targets the the lack of systematic observability and debugging capabilities for LLM-based applications which I think is an important need in the AI development ecosystem
- The MCP-based architecture for unifying metrics, control, and prompt management is well-articulated and could provide an interesting direction for future tooling.



## Weaknesses
- My main concern is that this paper explicitly states it presents "no benchmark experiments or claim specific performance improvements." For a systems paper proposing new development workflows, this is in my opinion critical. Without any evidence on the impact of proposed design, the validness of claims and contribution remains speculative.
- It's not clear to me how the MCP-based approach differs from existing LLM observability platforms (for example LangSmith, etc.). The claimed benefits of "real-time feedback" and "unified interfaces" seem to be already available in some of the current tools.

---

### Review · Reviewer_5wUY · 2025-07-13

**Summary Of Contributions:**

The paper „Mind the Metrics: Patterns for Telemetry-Aware In-IDE AI Application Development using Model Context Protocol (MCP) suggest to use MCP servers as central storage for telemetry data. IDEs can hook into the MCP servers to enable developers to retrieve telemetry data via queries (pattern 1). The second pattern has the idea to allow CI pipelines to hook into the MCP server to run prompt optimizations. The third pattern is for fully automated agent supervision, where some agent has access to MCP servers to monitor systems in operation. Notably, the paper provides no implementations or empirical data, but rather simply presents this high-level idea.

**Audience:**

No

**Claims And Evidence:**

No

**Requested Changes:**

Based on the issues listed above, I cannot request any specific changes that would not fundamentally alter the work.

**Strengths And Weaknesses:**

I do not think that this paper provides a sufficient contribution for TMLR. The general concept, i.e., we can hook an IDE, CI system, or agent, into some server, whether MCP or any other logging server is hardly a new idea. Everything that the authors suggest that MCP would offer, I would argue can be achieved with any other server as well, including natural language queries (e.g., by simply generating database queries for other servers to retrieve the requested information). The patterns that are suggested by the authors are also not innovative. Indeed, the related work section is full of exactly these patterns. Furthermore, only one of the patterns is actually directly related to the paper title, i.e., the IDE integration. The other two patterns are for CI and monitoring, i.e., different components, typically outside the IDE.

If there would have been an implementation that highlights specific advantages of using MCP, this might have some merit as a paper. However, as the authors highlight in the introduction and conclusion, this paper is only about the general concept.

The direct consequence of this lack of implementation is also that there is no evidence regarding what the authors suggest: MCP-based solutions would be helpful and better than other solutions. Again, this is admitted by the authors who delegate this to future work.

---

### Decision · Action_Editor_VbLF · 2025-08-27

**Recommendation:** Reject

**Audience:**

Yes

**Audience Explanation:**

New software development workflow with the support of LLMs is relevant for machine learning practitioners in this community. It could also be a important scenario for AI agent research.

**Claims And Evidence:**

No

**Claims Explanation:**

This submission proposes three progressive design patterns for an AI application development paradigm that integrates telemetry through MCP. Some reviewers appreciate it as an interesting direction for future AI application development. However, all reviewers agree that this paper only provides high level ideas without a concrete implementation or any type of verification that the proposed paradigm would be better than other solutions.